# Navigating the Neurobiology of Parkinson’s: The Impact and Potential of α-Synuclein

**DOI:** 10.3390/biomedicines12092121

**Published:** 2024-09-18

**Authors:** Erlandas Paulėkas, Tadas Vanagas, Saulius Lagunavičius, Evelina Pajėdienė, Kęstutis Petrikonis, Daiva Rastenytė

**Affiliations:** Department of Neurology, Lithuanian University of Health Sciences Kaunas Clinics, LT-50161 Kaunas, Lithuania; tadas.vanagas@lsmu.lt (T.V.); saulius.lagunavicius@stud.lsmu.lt (S.L.); evelina.pajediene@lsmu.lt (E.P.); kestutis.petrikonis@lsmu.lt (K.P.); daiva.rastenyte@lsmu.lt (D.R.)

**Keywords:** α-synuclein, αSyn, synucleinopathy, proteinopathy, Parkinson’s disease, etiology, pathogenesis, neurobiology, Lewy bodies, neurodegenerative diseases

## Abstract

Parkinson’s disease (PD) is the second most prevalent neurodegenerative disease worldwide; therefore, since its initial description, significant progress has been made, yet a mystery remains regarding its pathogenesis and elusive root cause. The widespread distribution of pathological α-synuclein (αSyn) aggregates throughout the body raises inquiries regarding the etiology, which has prompted several hypotheses, with the most prominent one being αSyn-associated proteinopathy. The identification of αSyn protein within Lewy bodies, coupled with genetic evidence linking αSyn locus duplication, triplication, as well as point mutations to familial Parkinson’s disease, has underscored the significance of αSyn in initiating and propagating Lewy body pathology throughout the brain. In monogenic and sporadic PD, the presence of early inflammation and synaptic dysfunction leads to αSyn aggregation and neuronal death through mitochondrial, lysosomal, and endosomal functional impairment. However, much remains to be understood about αSyn pathogenesis, which is heavily grounded in biomarkers and treatment strategies. In this review, we provide emerging new evidence on the current knowledge about αSyn’s pathophysiological impact on PD, and its presumable role as a specific disease biomarker or main target of disease-modifying therapies, highlighting that this understanding today offers the best potential of disease-modifying therapy in the near future.

## 1. Introduction

Parkinson’s disease (PD) is the second most prevalent progressive neurodegenerative disorder worldwide, yet it has remained a mystery for over 200 years. From its initial description in 1817 by English surgeon James Parkinson as “Shaking palsy”, to the current understanding based on typical Parkinsonian motor features of asymmetric bradykinesia, rigidity, tremor, and gait disturbances, significant progress has been made in understanding the disease’s pathophysiology and management, but its root cause remains elusive [1]. Various hypotheses have been proposed and nowadays the most prominent is that its neuronal loss is associated with the formation of α-synuclein (αSyn) aggregates in neurons of the substantia nigra, known as Lewy bodies (LBs) and Lewy neurites, leading to a perception of it being αSyn-associated proteinopathy and implicating αSyn as a central player in the pathogenesis of PD [2]. Following the identification of αSyn gene mutations in several Italian and Greek families affected by PD, αSyn was recognized as an important part in the pathophysiology of PD [3], shifting the paradigm from proteinopathy to proteinopenia [4]. Correspondingly, αSyn has been attributed to many functions in presynaptic, postsynaptic, and dopaminergic neurotransmission, and has a role in neurodegeneration and neuroinflammatory processes leading to neuronal loss. The identification of αSyn aggregation within LBs in patients, along with genetic evidence linking αSyn locus duplication and triplication, as well as αSyn point mutations, to familial Parkinson’s disease, has strengthened the importance of αSyn aggregation in the initiation and dissemination of LBs pathology throughout the brain [5].

Nevertheless, it is important to mention that abnormal misfolding and aggregation of αSyn in neurons and/or glia is a biological hallmark sign of PD as well as other synucleinopathies such as Lewy body dementia (LBD) and multiple system atrophy (MSA) [1]. Furthermore, non-motor and motor clinical symptoms of these diseases are similar, can overlap, especially in early stages, and they share similar not only clinical features, but neurochemical and morphological features too. In addition, there is an absence of indicators suggesting alternative diagnoses, such as atypical or secondary Parkinsonian syndromes. All these aspects make accurate diagnosis challenging, resulting in suboptimal clinical diagnostic accuracy even when the disease is clinically fully manifested [2]. Therefore, making an early diagnosis of Parkinson’s disease is a challenge not just for a general neurologist but also for specialists in movement disorders.

Notwithstanding, studies regarding αSyn aggregation assays are still highly promising in the detection of αSyn aggregates as a biomarker in PD. In vitro models of the seeding amplification assay (SAA) include real-time quaking-induced conversion (RT-QuIC) and protein misfolding cyclic amplification (PMCA). Using these methods, a patient-derived sample is incubated with a large amount of recombinant normal proteins, and repeated cycles of fragmentation and elongation are conducted to amplify the misfolded prion protein (PrPSc), which is detected by fluorescent dye. Research on αSyn aggregation biomarkers has focused on cerebrospinal fluid (CSF) αSyn seed amplification assays (SAAs), which are highly effective in distinguishing Parkinson’s disease (PD) from healthy controls [6,7,8]. Recent studies are exploring less invasive detection methods, including assays using blood serum and skin biopsies [9,10,11,12].

In 2024, new diagnostic criteria for PD were proposed for research purposes. Tanya Simuni and colleagues, as well as Günter Höglinger and colleagues, presented the two new PD biological frameworks: the neuronal αSyn integrated staging system (NSD-ISS) [13] and the SynNeurGe criteria [14]. The purposes of these frameworks are similar, and both have anticipated the classification systems that define disease subtypes, and the main attention is focused on the biological definition of PD, including αSyn identification. Moreover, both frameworks include the diagnostic significance of the earliest phases, the premotor and prodromal PD stages, based on genetic risks, the presence of pathological αSyn, and the loss of dopaminergic neurons. A breakthrough was achieved in 2023 by Okuzami and colleagues who presented blood-based assay targeting αSyn seeds in serum. The investigators showed high diagnostic performance in distinguishing diagnosed PD and MSA patients from non-neurodegenerative controls [9]. This research has implications regarding the identification of the prodromal phase of synocleinopathies. These significant data and the implementation of these two new frameworks based on biological features allow for the diagnosis of the disease even in asymptomatic individuals. This new conceptualization and these proposals open new opportunities, especially in the research area for redesigned trials for pathogenetic treatment that delays or prevents the disease.

In this review, we firstly provide emerging new evidence on the current knowledge on αSyn physiology in cellular structures and how pathological αSyn acts on dopaminergic transmission; secondly, we present the role of αSyn in neuroinflammation and discuss it as the important factor leading to neuronal death; thirdly, we demonstrate updated evidence about αSyn as a biomarker, and measure total αSyn and other species of αSyn in different biological fluids of PD initiation and progression; and subsequently, we shortly summarize disease-modifying therapies targeting αSyn for Parkinson’s disease.

## 2. Clinical Spectrum of Parkinson’s Disease

The Movement Disorder Society (MDS) clinical diagnostic criteria for Parkinson’s disease, established in the year 2015, are the most widely used tool for clinical and research purposes to enhance diagnostic accuracy [15]. Parkinson’s disease is characterized by classic motor symptoms, usually manifesting unilaterally or at least asymmetrically in addition to changes in posture and gait [16], and is associated with a variety of non-motor symptoms (NMSs), such as hyposmia, constipation, urinary dysfunction, rapid eye movement (REM) sleep behavior disorder (RBD), orthostatic hypotension, depression and anxiety, pain, and others [5]. NMSs are probably not associated with nigral degeneration and dopamine depletion, but indicate the neurodegeneration process of other structures, including the autonomic nervous system (ANS), both sympathetic and parasympathetic, and multiple other organs, especially the skin, colon, and salivary glands. These organs are affected by αSyn pathologic aggregates and might initiate spread via autonomic connectome to the CNS. These findings demonstrate that Parkinson’s disease is a multisystem disease [17]. Non-motor symptoms are frequent at early stages of diseases and can even precede motor deficits, sometimes for several years or even decades. These symptoms are defined as the prodromal disease stage. On the other hand, they are important in advanced Parkinson’s diseases and increase in severity with disease duration [2,18].

Increasing evidence suggests that prodromal and clinical Parkinson’s disease is heterogenous and can be classified into subtypes with different clinical manifestations, rates of progression, treatment responses, pathological mechanisms, and patterns [19]. Some features, such as RBD, early cognitive deficits, and autonomic dysfunction are associated with faster progression and a shorter life expectancy [19]. The slowest progression was seen in patients presenting with predominant motor features of mild severity. Additionally, genetically defined forms of the disease can differ from sporadic forms [2]. These findings suggest that perhaps soon Parkinson’s disease will not be characterized as a single nosological entity. Nowadays, novel tissue and fluid markers are under investigation and Parkinson’s disease is evolving from a clinical to a biomarker-supported diagnosis.

## 3. Alpha-Synuclein Plays a Central Role in the Pathogenesis of PD

Autopsy and in vivo studies have consistently identified pathological aggregates of misfolded αSyn in the form of Lewy bodies (LBs) in the brains of PD patients [20], as well as in biological fluids such as cerebrospinal fluid (CSF), blood, and peripheral tissues including the skin, gastrointestinal tract, and submandibular gland [21,22]. Abnormal forms of αSyn have been implicated in neuronal demise not only in PD but also in other synucleinopathies such as Lewy body dementia (LBD), multiple system atrophy (MSA), pure autonomic failure (PAF), and REM sleep behavior disorder (RBD). The widespread distribution of pathological αSyn aggregates throughout the body raises inquiries regarding the etiology of these pathologies, which has prompted the formulation of several hypotheses.

Firstly, the dual-hit hypothesis proposed by Braak posits that PD develops in response to unknown neurotropic agents entering the human body through the nasal and gastrointestinal routes, thereby initiating the formation of Lewy bodies [23]. Braak’s hypothesis has been revisited recently, suggesting that synucleinopathy (PD or LBD) may originate either in the olfactory bulb or the enteric nervous system, albeit rarely simultaneously [24]. Moreover, an alternative proposition known as the threshold theory suggests that PD arises concomitantly in the central and peripheral nervous systems, with motor symptoms delayed due to the larger functional reserve of midbrain dopamine and integrated basal ganglia systems, leading to the emergence of non-motor symptoms initially [25]. Conversely, the cortical pathogenic theory of PD posits that corticostriatal activity is harmful for nigrostriatal neurons by promoting the secretion of αSyn at vulnerable dopaminergic synapses. However, none of these theories fully elucidate the underlying mechanisms driving the pathological aggregation of αSyn. Moreover, emerging evidence implicates other significant biological components in the pathogenesis of PD, including genetic mutations, mitochondrial impairment, and other factors such as lysosomal dysfunction, alterations in calcium homeostasis, which all contribute to αSyn misfolding, as well as neuroinflammatory processes, which, together with Lewy body formation, result in neuronal degeneration [26] (Figure 1).

## 4. Family of Synuclein Proteins

Three types of synucleins, namely α-synuclein (αSyn), β-synuclein (βSyn), and γ-synuclein (γSyn), comprise a family of soluble phospholipid-binding proteins primarily expressed in neural tissue. While αSyn has been extensively studied due to its central role in Parkinson’s disease pathogenesis, βSyn and γSyn have received less attention as their functions are not directly associated with Parkinson’s disease.

Betta-synuclein, a 134-amino acid protein encoded by the *SNCB* gene on chromosome 5q35.2, is expressed in the neocortex, striatum, hippocampus, thalamus, and cerebellum, but is not found in Lewy bodies [27]. In contrast to that, Hashimoto et al. found a protective effect of βSyn overexpressing neurons to the pesticide rotenone [28], while in the same year, 2004, Ohtake et al. reported two unrelated patients with LBD who had two different heterozygote mutations in the *SNCB* gene, postulating that the absence or dysfunction of βSyn may impair the normal inhibitory effect on the formation of toxic αSyn fibrils, thereby indirectly contributing to the pathogenesis of synuclein-related neurodegenerative diseases [29]. On the other hand, gamma-synuclein, a 127-amino acid protein, encoded by the *SNCG* gene on chromosome 10q23.2, is implicated in oncological diseases [30].

αSyn was the first one of the synuclein protein family to be described in 1988 by Maroteaux and colleagues, who localized it to the axon terminals of presynaptic neurons [31]. In addition to the central nervous system (CNS), expression of the 140-amino acid αSyn protein has been identified in peripheral tissues. These include the heart, skeletal muscle, and various organs of the gastrointestinal tract, such as the stomach, small intestine, and colon. Furthermore, α-synuclein has been observed in the skin, adrenal glands, and reproductive organs [32]. Initially, αSyn became the subject of significant interest following the identification of its encoding gene, *SNCA*, located on chromosome 4q22.1. Mutations in the *SNCA* gene have been demonstrated to be causative factors in Parkinson’s disease [3].

## 5. Normal and Pathological Structure of α-Syn

In a living organism, α-synuclein exists in equilibrium between soluble and membrane-bound states [33]. A monomeric form of αSyn located in the presynaptic axon terminal of the neurons comprises a structure of three main domains: the amphipathic lipid membrane-binding N-terminal region (amino acids 1–60), which is crucial for the capacity of αSyn to interact with membranes [34], the intermediate aggregation-prone hydrophobic NAC (non-Aβ-amyloid component) region (amino acids 61–95), and calcium ions binding and the chaperone-like acidic C-terminal region (amino acids 96–140), which regulates αSyn binding properties to synaptic vesicles [26,35].

It is debated whether, in physiological conditions, αSyn comprises a multimeric conformation when bounded to the membranes of synaptic vesicles [36] or occurs as a helically folded tetramer that resists aggregation [37]. During post-translational modifications (PTMs) in the pathogenesis of Parkinson’s disease (PD), modified or misfolded αSyn monomers may become susceptible to aggregating into intermediate oligomeric structures, which subsequently assemble into insoluble fibrillar aggregates [38]. The structural diversity of αSyn oligomers is vast, including disordered forms and β-sheet conformation [39], both of which contribute to the formation of Lewy bodies.

Interestingly, not only are LBs toxic to neurons, but so are other forms of αSyn that are generated during more than 300 different PTM processes [40], including phosphorylation, proteolysis, nitration, glycosylation, lipidation, acylation, oxidation, ubiquitination, SUMOylation, and N-terminal acetylation [41]. A Lewy body is an intracellular inclusion of more than 200 different disaggregated oligomeric proteins [42] that are debated to both induce mitochondrial dysfunction, oxidative stress, and cytoskeleton impairment, and represent a failure of the cell to clear the misfolded and dysfunctional proteins [43]. An essential part of LBs is the αSyn protein [44], which is found to be 90% phosphorylated (p-αSyn) at serine 129 amino acid [45], while in physiological conditions, less than 5% of soluble monomeric αSyns exist in phosphorylated form [26]. Furthermore, the phosphorylation of tyrosine 129 residue located within the lipid membrane-binding N-terminal region of αSyn is known to induce aggregation. This process also hampers interactions of αSyn with chaperone proteins and exacerbates neurotoxicity [46]. Besides phosphorylation, other PTMs also contribute to the formation of propagation-prone oligomeric forms of α-syn. SUMOylation enhances the accumulation of αSyn by antagonizing its ubiquitination and impeding its degradation process [47]. Between 10% and 30% of αSyn within Lewy bodies exhibits truncation either at the N-terminus or the C-terminus, which enhances the tendency of αSyn to aberrantly misfold into distinctively toxic fibrils with altered prion-like seeding capacity [48]. Conversely, other PTMs contribute to the stabilization of the soluble monomeric αSyn form. For instance, phosphorylation at tyrosine 125, 133, or 135 does not affect it or the function is unknown for the aggregation of αSyn [49]. Balana et al.’s in vitro mechanistic investigations suggest that a glycosylation form called the O-GlcNAc modification might change the interactome of αSyn fibrils in a manner that diminishes their seeding activity in vivo [50]. Non-enzymatic oxidative modifications, like nitration and oxidation, were also suggested to impact αSyn dimerization and oligomerization [51]. Given the variety of αSyn protein aggregation promoting and inhibiting processes, it remains unclear which combination of factors facilitates the initiation of Parkinson’s disease. It is believed that interaction among genetics, previous illnesses, and environmental factors contributes to the incidence of PD [52]. Genetic mutations and PTMs of αSyn precede the pathological aggregation of αSyn, while the impairment of ubiquitin-proteasome and autophagy-lysosomal systems contributes to increased production rather than the elimination of misfolded αSyn, which then tends to organize into soluble oligomers, insoluble fibrils, and Lewy bodies [53].

## 6. Genetic Factors Contributing to the Pathogenesis of PD

Heterozygote mutations of the *SNCA* gene are almost fully penetrant, causing neuronal αSyn disease (PD or LBD) [13]. These include seven *SNCA* gene missense mutations (A30P, E46K, H50Q, G51D, A53V, A53T, A53E) [54], duplications, triplications [55,56], and polymorphisms in the promoter region and a distal enhancer element [57,58], all of which affect αSyn translation. The A30P polymorphism mutation reduces α-helical propensity, whereas the E46K mutation enhances contacts between the N- and C-termini. Additionally, the A30P mutation favors αSyn oligomerization, while the A53T and E46K mutations promote fibrillation [59]. Several other genes have been clinically associated with the autosomal dominant inheritance pattern of PD—*UCHL1* (ubiquitin C-terminal hydrolase 1; PARK5), *LRRK2* (leucine-rich repeat kinase 2; PARK8), *GIGYF2* (GRB10 interacting GYF protein 2; PARK11), *VPS35* (vacuolar protein sorting 35; PARK17) [60,61,62], and *GBA1* (glucocerebrosidase 1). UCHL1 protein deficiency may disturb intracellular mitochondrial homeostasis because it has a crucial role in regulating mitophagy independently of the PINK1-Parkin pathway. A group of Republic of Korea scientists demonstrated that in UCHL1 knockout cells, cellular pyruvate production and ATP levels are diminished; therefore, the activity of AMP-activated protein kinase (AMPK) is highly induced, resulting in the disruption of glycolysis and induction of mitophagy [63]. GIGYF2 has been shown to play a role in the regulation of IGF-1R (insulin growth factor 1R) trafficking [64], suggesting insulin dysregulation as a disease-specific mechanism for both late-onset PD and cognitive dysfunction [62]. The role of insulin in the regulation of brain dopaminergic activity was proposed a decade ago, since the identification of elevated levels of IGF-1 and IGF binding proteins (IGFBPs) in the serum and cerebrospinal fluid (CSF) of patients with PD [65]. LRRK2 gene mutations are thought to cause pathological activation of LRRK2 kinase activity, which may induce oxidative stress, disrupt membranes and vesicle trafficking, and greatly affect endolysosomal function. Neuroinflammation and several therapeutic approaches, including antisense oligonucleotides, and kinase inhibitors targeting this gene, have been implicated in clinical trials [66]. The *VPS35* D620 mutation-caused protein deficiency has been suggested to increase αSyn accumulation in the brain via indirect impairment of a lysosomal protease Cathepsin D which aids in the degradation of lysosomal cargo, including αSyn [67]. The *GBA1* gene codes for a lysosomal enzyme called glucocerebrosidase, a defect which leads to accumulation of glucocerebroside and subsequent lysosomal dysfunction. Biallelic mutation of *GBA1* causes Gaucher disease, while heterozygote carriers of the *GBA1* mutation are estimated to have a more than five-fold increased risk of developing PD, which makes alteration of this gene the most important genetic risk factor [68].

The autosomal recessive (AR) inheritance pattern of PD is associated with the *Parkin* gene (*PARK2*), *PINK-1* gene (*PARK6*), and *DJ-1* gene (*PARK7*), while several other rare forms of PD include mutations in the *ATP13A2* gene (ATPase 13A2; PARK9), *FBXO7* gene (F box 7 protein; PARK15), *DNAJ6* gene (PARK19), *SYNJ1* gene (PARK20), and *VPS13C* gene (*PARK23*). AR-inherited forms of PD seem to be associated with accumulation mechanisms different than those of αSyn, including mitochondrial dysfunction, ROS generation, and inflammation promoting processes. The Parkin protein belongs to the ubiquitin-proteasome system, which mediates the targeting of the proteins for degradation, and has a role in the mediation of mitophagy and neuroinflammation. Detection of elevated cytokine levels suggests the possibility of anti-inflammatory treatment to modulate the progression of PD [69]. *PINK1*, in conjunction with *Parkin*, is also associated with mitochondrial metabolism. Throughout the process of mitochondrial degradation, these proteins mitigate the release of mitochondria-derived vesicles, thereby reducing reactive oxygen species (ROS) generation, mitochondrial antigen presentation, and the activation of adaptive immune responses [70]. *DJ-1*’s function is unclear, but it is thought to prevent oxidative stress. One study reveals that elevated expression levels of A53T αSyn are inversely associated with DJ-1 expression levels in A53T αSyn mice exposed to subtoxic doses of 1-methyl-4-phenyl-1,2,3,6-tetrahydropyridine (MPTP) [71], suggesting an interaction between DJ-1 and αSyn. This indicates that targeting DJ-1 could be pertinent in addressing this relationship [72].

## 7. Environmental and Other Factors Contributing to the Incidence of PD

Genetic mutations contribute only to 5–10% of all PD cases, and the rest of them are sporadic and thought to be influenced by a combination of environmental factors, certain lifestyles, and habits [40]. Age is the single most consistent risk factor for developing PD. From 1990 to 2019, there was a noticeable rise in the burden of Parkinson’s disease (PD) on a global scale, as well as in most regions and countries [73]. Parkinson’s disease impacts approximately 0.5–1% of the population aged between 65 and 69 years, with the prevalence rising to 1–3% among individuals aged over 80 years [74]. Sex is another nonmodifiable risk factor, meaning that men suffer from PD two times more often than women. Intoxication with pesticides, herbicides, heavy metals, and various solvents account for the next most influential risk factors for developing PD after age and sex. The Parkinson Environment Gene (PEG) study revealed that occupational exposure to carbamates was associated with a 455% increase in the risk of PD, while the use of organophosphorus (OP) and organochlorine (OC) pesticides doubled the risk. Additionally, the risk of PD increased by 110–211% with occupational exposure to fungicides, herbicides, and insecticides. Moreover, individuals using any pesticide occupationally for more than 10 years had twice the risk of PD compared to those with no occupational pesticide exposure [75]. Pesticide rotenone and herbicide paraquat have been shown to increase ROS generation, which results in the overexpression and aggregation of αSyn, causing damage to and death of dopaminergic neurons [76,77,78]. Solvents with trichloroethylene (TCE) or perchloroethylene (PCE), that are found in numerous consumer products for dry cleaning, as degreasing agents, in typewriter correction fluids, paint removers, and carpet cleaners, are also linked to PD. A twin study showed a five-fold increased chance of developing PD after exposure to TCE compared to the unexposed twin. Goldman and colleagues reported an interval from 10 to 40 years between exposure to TCE and the clinical symptoms of PD [79]. In rodent models, TCE and PCE are believed to exert their toxic activity by causing mitochondrial complex I dysfunction, increasing ROS generation, inducing neuroinflammation, and subsequently αSyn phosphorylation and accumulation in the substantia nigra and striatum [80]. Materials used in the electrical industry as coolants and insulators like polychlorinated biphenyls (PCBs) and carbon tetrachloride (CCl4) are implicated in the pathogenesis as well, since they are found in post-mortem brains of PD patients [81]. Exposure to heavy metals (Al, Fe, Cu, Mg, Hg, Pb, Mn, Zn, Bi, Tl) contributes to increased ROS generation, which then can directly or indirectly cause αSyn misfolding and aggregation due to oxidative modifications of amino acid side chains, induce DNA methylation, disrupt Ca^2+^ homeostasis, and cause mitochondrial dysfunction and dopamine synthesis impairment [82]. Additional risk factors for developing PD include head traumas which disrupt the blood–brain barrier and induce long lasting inflammation, living in rural areas and drinking well water which may be polluted with different toxic materials mentioned above, a higher body mass index, and specific diets (consuming more dairy products, eating animal fats, sea mammals, and whale meat) [83]. Previous illnesses like cancer (e.g., melanoma), hormonal changes during postmenopausal hormone distribution, and some autoimmune diseases, like diabetes mellitus, scleroderma, inflammatory bowel diseases, hypothyroidism, and hyperthyroidism, are also risk factors [84]. On the other hand, several environmental and medical factors possess protective features, probably by reducing inflammatory activity [83]. These include smoking, physical activity, drinking more tea and coffee, consuming vitamin E, polyunsaturated fatty acids, the Mediterranean diet, having increased estrogen levels (female sex), hyperuricemia, as well as using non-steroidal anti-inflammatory drugs (NSAIDs) like ibuprofen, calcium channel blockers, and statins [83].

## 8. Overview of the Proposed Functions of α-Syn

Parkinson’s disease, as well as other neurodegenerative diseases like Alzheimer’s disease (AD), Lewy body dementia (LBD), multiple system atrophy (MSA), progressive supranuclear palsy (PSP), and others have long been defined by the aggregation of pathological proteins which disrupt many functions of the nervous system. The gain-of-function (GOF) mechanism of the disease highlights the toxicity of aggregating proteins like amyloid-beta in AD, α-synuclein in PD, LBD, or MSA, and tau protein in PSP. However, important to note is the following loss-of-function (LOF) part of the mechanism, when consumed proteins stop performing their normal functions. A shift from a proteinopathy to a proteinopenia paradigm in neurodegenerative diseases [4] is important in PD since α-synuclein has been attributed to many functions in presynaptic, postsynaptic, and dopaminergic neurotransmission, and has role in neurodegeneration and neuroinflammatory processes [85] (Figure 2).

### 8.1. Role in the Presynaptic and Postsynaptic Neurotransmission Mechanisms

The exact role of αSyn is still under investigation but growing evidence shows that αSyn, which is mainly located in presynaptic terminals of neurons, is highly involved in endocytosis, exocytosis, stabilizing membranes, and regulating the trafficking of the vesicles [26]. Presynaptic nerve terminals release and reuptake neurotransmitters at high frequencies, which is mediated by the rapid assembly and disassembly of SNARE (soluble N-ethylmaleimide sensitive factor) complex proteins [86]. Alpha-synuclein has a role in promoting SNARE assembly during synaptic vesicle docking and fusion [87]. This protein acts as a SNARE complex chaperone by a non-enzymatic mechanism entailing the simultaneous binding to phospholipids through its N-terminus and to vesicle associated membrane protein 2 (VAMP2 or synaptobrevin-2) via its C-terminus [88]. Additional data indicate that αSyn acts together with VAMP2 and synaptic vesicle-attached synapsins, helping to maintain physiologic clustering of synaptic vesicles [89]. The essential role of the αSyn-VAMP2 bond is supported by the co-localization of the same pentapeptide region in αSyn and VAMP2 proteins that is recognized by chaperone-mediated autophagy (CMA) [90]. It is known that CMA is strongly involved in synucleinopathies [91]. The absence of normal levels of αSyn has been linked to a decrease in the quantity of synaptic vesicles in distal pools [92], whereas its excessive expression has been demonstrated to elevate the number of vesicles adhered to the membranes [93]. Moreover, all types of synucleins, namely αSyn, βSyn, and γSyn, have been shown to be essential for the swift internalization of synaptic vesicle membranes, as the lack of them corresponded with slowed endocytosis [94]. Also, in a study with triple knockout (TKO) mice devoid of αβγ-synucleins, the mice displayed age-dependent neurological impairments, manifested reduced SNARE complex assembly, and died prematurely. Hence, synucleins likely play a role in maintaining the normal assembly of the SNARE complex within presynaptic terminals during aging [88]. Additionally, αSyn inclusions are detected within the presynaptic active zone, concomitant with a decline in active zone protein, and diminished endocytic retrieval of synaptic vesicle membranes during vesicle recycling, causing early neuronal dysfunction [95]. An alternative hypothesis also shed light on αSyn impact on lipid metabolism which are important for synaptic localization, vesicle cycling and modulation of synaptic integrity [96]. Cholesterol facilitates interactions between αSyn oligomers [85], which might explain the protective effects of statins.

Besides the presynaptic role of αSyn, new investigations suggest the postsynaptic involvement of native and pathogenic forms of αSyn, which have a role in modulating glutamatergic ionotropic and metabotropic receptor activity, glutamatergic activity of astrocytes, and striatal synaptic plasticity [85]. Synaptopathy induced by preformed fibrils (PFF) of αSyn involves the dysfunction of N-methyl-D-aspartate (NMDA) receptors. In an investigation utilizing post-mortem human brain tissue, levels of αSyn monomers and oligomers were observed to selectively rise with age in the striatum and hippocampus, which correlated with the underexpression of the NMDAR GluN1 subunit [97]. On the other hand, accumulation of the glycated form of αSyn was demonstrated to cause overexpression of glutamate-related proteins (α-amino3-hydroxy-5-methyl-4-isoxazolepropionic acid (AMPA) receptors, glutaminase, vesicular glutamate transporter (VGLUT), excitatory amino acid transporter 1 (EAAT1), and NMDA receptors) in the midbrains of mice [98]. Interestingly, the modulation of glutamate overexpression by, e.g., NMDAR antagonists like amantadine, is beneficial for treating levodopa-induced dyskinesias [99]. Moreover, glycation is associated with motor, cognitive, and olfactory dysfunction [85], which is very common in PD, suggesting that anti-diabetic medications like GLP-1 (glucagon-like peptide 1) antagonists can help modify the neurodegenerative course of PD [100]. Oligomers of αSyn have been also shown to disturb astrocyte function by inducing the calcium-dependent release of glutamate, and overactivation of extracellular NMDAR, thus contributing to synapse loss [101]. Recently, alterations in striatal synaptic plasticity have been suggested to occur in response to the toxic effects of misfolded αSyn. Previous ex vivo studies showed that induced overexpression of human αSyn in the substantia nigra by viral means resulted in reduced striatal dopamine transporter (DAT) levels, impaired motor learning, and hindered learning-induced long-term potentiation (LTP) prior to the onset of dopaminergic neuronal loss [102]. Later in vivo studies conducted by the same group of researchers reported that animals injected with αSyn-PFF displayed anxiety-like behavior and hypokinesia [85]. These alterations in neuronal function within the substantia nigra pars compacta (SNpc) and striatum, along with the behavioral dysfunctions, were noted prior to the onset of apparent neuronal death [103]. Additionally, it selectively impedes the induction of long-term potentiation (LTP) in striatal cholinergic interneurons (ChIs), resulting in early alterations in memory and motor function [104]. These results suggested the emergence of network dysfunction preceding neurodegeneration [85].

### 8.2. Role in Dopaminergic Neurotransmission

The presence of intracellular pathological inclusions containing the αSyn protein within dopaminergic neurons is a hallmark feature of Parkinson’s disease (PD). However, the precise mechanisms by which αSyn contributes to the vulnerability of dopaminergic neurons remain elusive. Also, the inability to access diseased tissue has posed a limitation in studying the progression of pathophysiology preceding the degeneration of dopamine neurons. Native αSyn is present in presynaptic terminals of neuronal cells [31], mitochondria [105], the endoplasmic reticulum [106], and the Golgi apparatus [107]. Accumulation of misfolded oligomers or PFF of αSyn contributes not only to presynaptic and postsynaptic dysfunction discussed above, but also disrupts the mitochondrial function of electron transport chain’s complex I [108], the ubiquitin-proteasome system [109]. However, not every part of the central nervous system is affected by synucleinopathies. Dopamine (DA) neurons are argued to be the most vulnerable because of their specific structural and functional characteristics. DA neurons are characterized by extensively branched axons, a high density of neurotransmitter release sites, and unique pacemaking activity. Pacemaking activity is sustained by the rhythmic influx of calcium ions (Ca^2+^) into the cell. While the entry of Ca^2+^ stimulates energy (ATP) production within the neuron, the extrusion of Ca^2+^ depletes the generated energy. Substantia nigra pars compacta dopaminergic neurons are believed to operate at a delicate equilibrium, making them very susceptible to any perturbation between energy demand and production [110].

Under physiological conditions, αSyn acts to ensure the proper storage of DA by regulating the activity of vesicular monoamine transporter 2 (VMAT2). By maintaining high VMAT2 activity, αSyn may contribute to the protection of dopaminergic neurons from cell death, as the storage of DA within synaptic vesicles also serves to shield it from oxidative damage [111]. Exposure of neurons to αSyn oligomers leads to an elevation in intracellular Ca^2+^ levels, subsequently triggering DA release in striatal neurons [112]. Oscillations of Ca^2+^ release also induce a Ca^2+^ influx into the mitochondria, initiating oxidative phosphorylation and ATP production, which in turn promote a toxic cascade of further aggregating αSyn [113]. Animals models support these hypotheses, as the overexpression of αSyn is concomitant with a decrease in tyrosine hydroxylase, an enzyme involved in the production of DA, and an overall decrease in DA in the striatum [112].

### 8.3. Inflammation and αSyn

Increasing evidence links PD to inflammatory processes both in the periphery via gut–brain dysbiosis and the central nervous system (CNS). Evident is the prodromal phase of PD, which clinically involves a range of non-motor symptoms (NMSs) preceding motor decline in the later stages. Common NMSs include dysautonomia characterized by gastrointestinal dysfunction like excessive drooling, obstipation, early satiety feeling, and bladder dysfunction with urinary incontinence or retention. Together with olfactory and sleep disturbances, these symptoms support Braak’s theory where Lewy bodies are believed to form in the periphery and spread in a caudal–rostral pattern to the brainstem and higher brain centers [23]. During the prodromal phase, the combination of toxic, epigenetic, aging, environmental, and immunological factors can set the stage for the ideal circumstances to start Parkinson’s disease. However, the impact of age on the declining performance of our immune system is underappreciated. It could be postulated that all the toxic, environmental factors that the human body faces during its lifetime can be overcome by the intact immune system, but at certain point in age, the defense mechanisms become insufficient. This process called immunosenescence is characterized by age-acquired immunodeficiency and inflammaging. Inflammaging is characterized by decreasing levels of circulating cytokines like C-reactive protein (CRP), tumor necrosis factor (TNF), or interleukin 6 (IL-6) [114]. Growing evidence shows that alterations of the innate and adaptive immune system contribute to developing PD. If we think about Braak’s theory, the mechanism of immune system dysfunction suggests that intestinal dysbiosis increases levels of circulating pro-inflammatory cytokines, thus activating innate and adaptive immune cells (T-cells, macrophages, monocytes) which infiltrate the CNS across the blood–brain barrier and induce microglia activation and chronic inflammation [115]. But what does αSyn have to do with neuroinflammation?

The main immune cells of the CNS are microglia. Physiologically glial cells are supposed to express low levels of αSyn, since it is mainly located on the presynaptic terminals of neurons. However, a recent study shows that, if neurons are damaged, the released αSyn induces the activation of microglia, which then engulf αSyn into autophagosomes for degradation through selective autophagy, commonly referred to as synucleinphagy [116]. Microglial activation is also supported by in vivo studies with PET ligand [^11^C]-(R)-PK11195 tracing activated microglia cells in PD models [117]. Accumulation of αSyn aggregates induces activation of inflammasomes, particularly nucleotide-binding oligomerization domain-leucine-rich repeat-pyrin domain-containing 3 (NRLP3), which exacerbates inflammation in the CNS by secreting cytokines IL-18, IL-1β [118]. Increased production of various types of cytokines causes chronic inflammation, which correlates with neuronal damage and the further release of αSyn, creating a vicious cycle of propagating and accumulating αSyn. Accordingly, a recent systemic review and meta-analysis demonstrated significantly increased levels of IL-6, TNF-α, IL-1β, MCP-1 (monocyte chemoattractant protein 1), and CRP in both peripheral blood and CSF, as well as increased levels in IL-4, IFN-γ, STNFR1 (soluble tumor necrosis factor receptor 1), and fractalkine only in blood samples of PD patients [119]. These findings provide further clinical evidence that PD is associated with a distinct peripheral and central inflammatory response. Moreover, the previously cited study showed that the impairment of microglial autophagy in mice expressing human αSyn led to the buildup of misfolded αSyn and triggered the degeneration of midbrain dopaminergic neurons, identifying the protective role of microglia [116]. Evidence that the autophagy–lysosome system is one of the crucial protective mechanisms protecting individuals from developing synucleinopathy is further supported by human genomic studies which reveal that mutations in the *PINK1*, *Parkin*, *GBA*, and *LRRK2* genes, which are causative of PD, also disrupt lysosomal functions [120]. Additionally, to microglia, the innate immune system, which provides first-line defense against pathogens and insults like αSyn in PD, includes macrophages and monocytes, while there is a lack of evidence of the involvement of neutrophils and dendritic cells [121]. In PD patients and mouse models, monocytes expressed toll-like receptor (TLR) 2 and 4 [122], where TLR4 correlated with the extent of immune activation in the substantia nigra (SN), as assessed by a PET scan [123]. Experimental data show that macrophages can also react to both naturally produced αSyn and αSyn found in the extracellular surroundings through TLR2 or TLR4 [124].

Furthermore, the adaptive immune response has an equally important role in targeting αSyn during PD. One of the first pieces of evidence of neuroinflammatory pathogenesis was presented in 1988 by McGeer et al., who demonstrated the presence of HLA-DR+ reactive microglia and CD3+ T-cell infiltrates in post-mortem tissue obtained from patients with PD [125]. The adaptive immune system provides long-term memory for a given pathogen and comprises T-cells, B-cells, and natural killers (NKs). Considering the response of NK cells to αSyn, their presence in the SN indicates their potential readiness to react to both monomeric and aggregated forms of αSyn, suggesting an early protective effect [126]. B lymphocytes produce antibodies that target specific pathogens. One study with 25 PD patients identified 10 unique IgG antibodies against αSyn, 3 of which were demonstrated to effectively inhibit αSyn aggregation [127]. Moreover, autoantibodies against αSyn, that have been found in plasma and CSF, were demonstrated to correlate with disease severity and serve as a potential biomarker for PD [128]. One study involving T lymphocyte-mediated immunity suggested there is a diminished ratio of total T-cells and CD4+/CD8+ ratio in PD patients [129] and this can lead to enhanced killing of the cells [121]. Moreover, effector T cells isolated from PD patients demonstrate elevated production of proinflammatory cytokines when exposed to T cell activators in vitro [130].

## 9. Fluid-Tissue Alpha-Synuclein Biomarkers

Early studies focused on measuring the values of different αSyn species in CSF and blood. Three main forms of αSyn (total, oligomeric, and phosphorylated αSyn) were assessed using various immunoassays such as ELISA, Western blot, and Luminex assay. At first, studies showed that total αSyn in CSF was lower in PD patients compared to healthy controls [131]. However, further trials found severe limitations in its use as total αSyn values varied greatly among different studies. Preanalytical reasons such as accidental CSF sample contamination with blood meant significantly increased total αSyn values. Research has shown that αSyn is abundantly found in red blood cells and is released in soluble-free form during hemolysis. A study by Kruse et al. showed that absolute values of total CSF αSyn varied greatly between laboratories even when the same samples and same lots of assays were applied [132]. The remaining two species of αSyn are deemed to be pathological. Oligomeric αSyn form is the predecessor of stable amyloid fibrils found in Lewy bodies. Autopsy studies have shown that only a small proportion of αSyn is phosphorylated in healthy controls (< 4%), while more than 90% of αSyn found in Lewy bodies is in phosphorylated form (pS129) [133]. Due to its extremely low extracellular concentration, an accurate and reliable measurement of pS129 quantity is complicated [134]. Multiple meta-analyses found that measuring the absolute value of αSyn species had inadequate diagnostic accuracy differentiating PD patients from control groups [131,135,136,137,138].

Further research led into the exploration of peripheral αSyn deposits. Post-mortem studies have shown no concrete evidence of incidental phosphorylated αSyn deposits in autonomic nerves without underlying brain pathology. Based on that knowledge, Ruffman et al. carried out a study examining the detection of αSyn in gastrointestinal mucosa using immunohistochemistry (IHC). The results showed overall sensitivity was 61% and specificity was 67%. Comparison of multiple studies had shown low replicability with results ranging widely [139]. Based on the same principle, numerous research groups published more promising results using skin punch biopsy immunofluorescence (IF). As it was a less invasive and cheaper method, a great number of studies tried to replicate the results. One of the first discoveries made was that different skin biopsy sites have varying levels of sensitivity and specificity, with the paracervical vertebral region and distal leg sites (inner thigh or lower calf) showing the best reproducibility [10]. Various other preanalytical factors such as the fixation method and biopsy thickness influenced the results as well. To standardize these factors, a multi-center, prospective, blinded study was conducted [11]. The Synuclein-One study included 343 individuals, of whom 223 met the clinical criteria for a synucleinopathy, and 120 were in the control group with the goal of clarifying the accuracy of skin punch biopsy IF in detecting p-αSyn deposits. The results published in March 2024 show that cutaneous p-αSyn deposits were correctly identified in 95.5% of all synucleinopathy patients, with the PD group having a 92.7% p-αSyn-positive rate. Moreover, researchers found that the numerical measure of p-αSyn correlated with MDS-UPDRS III total score, H&Y score, and other scales depicting disease severity [12]. This could prove to be of great importance as the skin punch biopsy can be repeated any number of times and would be an instrumental tool in longitudinal studies.

In 1994, Kocisko et al. described how a pathological, misfolded prion protein (PrPSc) interacted with alpha-helix rich protein (PrPc), turning it into a beta-structure-rich insoluble protein. Due to the prion-like properties of αSyn, a seed amplification assay (SAA) can be used to identify misfolded αSyn [140]. In vitro models of the SAA include real-time quaking-induced conversion (RT-QuIC) and protein misfolding cyclic amplification (PMCA). Using these methods, a patient-derived sample is incubated with a large amount of recombinant normal proteins, and repeated cycles of fragmentation and elongation are conducted to amplify PrPSc, which is detected by fluorescent dye thioflavin T. When a sample reaches the fluorescence threshold, it is deemed RT-QuiC-positive. Thus, SAA is a qualitative method that has a binary output [6].

Due to previous experience with αSyn species and CSF, most of the research initially focused on the CSF αSyn SAA. A 2023 meta-analysis comparing 26 αSyn CSF SAA studies (2025 patients) showed a high pooled sensitivity of 91% and a specificity of 95% when distinguishing PD patients from healthy controls. Eleven studies used the CSF αSyn SAA to distinguish PD and MSA patients; their results showed a pooled sensitivity of 91% and pooled specificity of only 50% [7].

Even though the results of the CSF αSyn SAA are promising, researchers are again exploring less invasive methods to detect pathological αSyn. Okuzumi et al. published a protocol using a modified assay system called immunoprecipitation-based real-time quaking-induced conversion (IP/RT-QuIC), which enabled the detection of pathological αSyn in the serum of patients with synucleinopathies. The reported sensitivity and specificity of distinguishing PD patients from healthy controls were 94.6% and 92.1%, respectively. The group used transmission electron microscopy (TEM) to compare αSyn fibrils derived from CSF and serum and found no morphological difference. However, no other groups have replicated the results using the IP/RT-QuIC system and further studies are needed [9].

Multiple studies have shown that the SAA works in detecting αSyn not only in CSF and blood serum, but also in the gastrointestinal tract, salivary glands, and skin. Zheng et al. found the skin punch biopsy αSyn SAA to have the second highest pooled sensitivity and specificity after CSF [7].

SAA results vary when patients are categorized by the genetic form of the disease. For example, Siderowf et al. found that the sensitivity of the CSF αSyn SAA for *GBA* carriers was 95.9% compared to 67.5% for *LRRK2* carriers. Similar results are reported using other biofluid biomarkers, i.e., Okuzumi et al. found that all PRKN-positive PD patients had negative IP/RT-QuIC results. These results correspond to autopsy studies done on LRRK2-PD and PRKN-PD gene carriers, in which αSyn pathology is found in less than half of these patients [8].

Isolated rapid eye movement (REM) sleep behavior disorder (RBD) and hyposmia are the main prodromal symptoms of synucleinopathies. Studies have shown the 10-year phenoconversion rate of RBD to be as high as 67.5% [141]. Siderowf et al. showed the ability to detect pathological αSyn using the CSF SAA in the prodromal stages of the disease. The study examined 1123 patients from the Parkinson’s Progression Markers Initiative (PPMI) cohort and reported the sensitivity of the αSyn SAA for detecting all PD cases to be 87.7%. When evaluating patients with RBD, 84.8% tested positive for the αSyn SAA. Similar results were reported in the hyposmia groups, with 16 of 18 cases being positive for the αSyn SAA. 86% of individuals with RBD and hyposmia had positive αSyn SAA results [8].

Studies that have assessed the diagnostic performance of the SAA in distinguishing Parkinson’s disease (PD) and other synucleinopathies from healthy controls have been compiled in a summary table (Table 1) [8,9,142,143,144,145,146,147,148,149,150,151,152,153,154]. This table includes key characteristics, offering a concise comparison that highlights the variability and diagnostic potential of these biomarkers across different contexts.

Although the SAA can detect pathological αSyn in prodromal cases, the exact time frame and rate of phenoconversion are unknown. This raises the ethical concern of establishing a diagnosis. Past years have shown that the ‘labeling of pre-patients’ has been and will only become more prevalent in the field of neurodegenerative diseases. As this causes a paradigm shift from late clinical diagnosis to patients without any clinical symptoms, the field has displayed an urgent need for studies on risk disclosure. Recent papers based on expert opinion recommend exercising caution and respecting patients’ autonomy and right to know or not to know [155]. Currently, there is no available causal treatment for Parkinson’s disease, with many studies underway; thus, it is beneficial to have set the inclusion criteria for prodromal patients including biomarkers. However, this could place an exhausting psychological burden on the patients as it could lead to stigmatization, anxiety, and depression. Due to the chance of false positives, it may result in some individuals never developing synucleinopathy but living their life believing they inevitably will.

Even though the SAA has often been referred to as a breakthrough, concerns need to be raised. The SAA requires vast funds and personnel expertise, which may hold back its availability. It is also limited in its differential diagnostic value. Due to the high incidence rate of synucleinopathy comorbidity, as many as 45% of AD patients are CSF αSyn SAA-positive, with positive cases more often presenting atypical forms [156]. The biofluid αSyn SAA also has an unsatisfactory sensitivity and specificity for certain gene carriers who have clinically confirmed PD. Thus, based solely on the SAA results, a synucleinopathy diagnosis cannot be established, as further neuro clinical examination and imagining are needed.

## 10. Alpha-Synuclein as a Therapeutic Target

Despite extensive achievements managing symptoms of Parkinson’s disease, there is no approved disease-modifying treatment to stop or slow the progression of neurodegeneration yet. Therapeutic targets include α-synuclein (αSyn), neuroinflammation, addressing mitochondrial function, counteracting dysfunction of *GBA* and *LRRK2* genes, and the application of regenerative and restorative therapies.

The biggest number of clinical trials is dedicated to targeting αSyn. Pathology lies in the abnormal misfolding, propagation, and aggregation of αSyn. Throughout this process, αSyn undergoes structural alterations, transitioning from its intrinsically disordered state to a more organized β-sheet structure. While both conformations have been associated with cytotoxicity, recent perspectives suggest that oligomers, rather than fibrils, primarily possess this property [157]. Various strategies are under investigation in the field, including reducing the expression of αSyn (e.g., through RNA interference [158]), inhibiting fibrillation with small molecules [159], and enhancing cellular degradation pathways such as autophagy (including chaperone-mediated autophagy or macro-autophagy) or proteasomal clearance [160]. Immunotherapy presents an appealing option, as antibodies targeting αSyn could potentially disrupt multiple processes implicated in PD pathogenesis. These antibodies might prevent the formation of pathogenic species, facilitate the clearance of existing species, and shield pathogenic forms to prevent neurotoxicity and inflammatory responses. The main approaches are passive and active immunization.

Passive immunization involves administering anti-αSyn monoclonal antibodies targeting specific epitopes of either the N- or C-terminal to lower αSyn concentrations and block cell-to-cell transmission. Cinpanemab/BIIB054 was one of the first monoclonal antibodies developed which binds to the N-terminal and forms plasma complexes with αSyn. However, clinical study was terminated in the phase II SPARK clinical trial as measures of PD progression (MDS-UPDRS score, DaT-SPECT) did not differ from the placebo group over a 52-week period [161]. Another monoclonal antibody, namely Prasinezumab/PRX002, which targets the C-terminal, was showing better results in the phase II PASADENA trial as it reduced the time to progression of motor decline measured by MDS-UPDRS III by 5 points, but a recent update in 2022 revealed that it had no meaningful effect on global or imaging measures of PD progression as compared with placebo and was associated with infusion reactions [162]. Negative results from these two trials targeting αSyn therapies are disappointing, but do not mean that targeting αSyn is an ineffective approach. Nevertheless, several limitations regarding these two trials are important. First of all, it is unclear whether sufficient antibodies enter the brain to have a therapeutic effect, despite the presumption based on data from phase I studies [163]. Second, from phase I trials, it was unknown how antibodies bind to αSyn in the CSF and brain, not only in the serum. Furthermore, it is unclear if the MDS-UPDRS scale is a sufficient tool to detect treatment efficacy in phase II trials; maybe a more sensitive and accurate outcome measure instrument is needed, such as longer assessments in the participants’ regular environment with an accelerometer, actigraph, or other digital measures. PD progresses slowly, especially in the early stages of the disease in which these studies were conducted, so longer trial periods might be beneficial too [163].

Several additional candidates for passive immunization are being investigated with MSA patients. MEDI1341 from AstraZeneca was demonstrated in preclinical studies to rapidly enter the CNS, lower free extracellular αSyn levels in the interstitial fluid (ISF), CSF compartments, and reduce αSyn accumulation and propagation along axons, and is now being investigated in a phase II clinical trial [164]. Similarly, the Lu AF82422 monoclonal antibody from Lundbeck A/S completed a phase I trial where it proved to be appropriate for further clinical development, and is now continued in a phase II trial also with MSA patients [165].

Active immunization enables the individual’s own immune system to recognize and target specific unfavorable antigens. Active immunization is usually less frequently administered and thus more cost efficient than passive immunization. Nevertheless, there is no guarantee that sufficient immunological memory will form after active immunization, considering that PD patients are usually older with diminished humoral and cellular responses due to imunosenescence [115]. Active immunization against αSyn has been performed in several clinical trials with Affitope PD01A and Affitope PD03A by AFFiRiS in PD and MSA and UB-312 in PD by United Biomedical Inc., United Neuroscience Ltd. [166]. Phase I trials assessed the safety and tolerability of PD01A and PD03A in patients with Parkinson’s disease (PD) and multiple system atrophy (MSA) [166]. Both active treatments elicited a persistent IgG antibody response against the immunization peptides, although PD01A showed greater results over PD013 in terms of seroconversion rates. In July 2021, a phase II trial launched with a new optimized PD01A formula, called AC-7104 [166].

## 11. Discussion

PD remains one of the most complex neurodegenerative disorders, where the role of αSyn in its pathogenesis has been a focal point of research. This review contributes to the existing body of literature by consolidating the latest findings on αSyn’s role in PD pathogenesis, diagnostic biomarkers, and therapeutic strategies. By highlighting both the physiological and pathological aspects of αSyn, we provide a more comprehensive understanding of its role in PD.

An examination of the pathogenesis of Parkinson’s disease (PD) reveals that alpha-synuclein (αSyn) serves as a critical link among the numerous impaired mechanisms observed in this disorder. Both genetic and environmental factors contribute to a wide array of post-translational modifications of αSyn, resulting in the misfolding and aggregation of this protein. Failure of protein elimination systems allows αSyn aggregates to accumulate in both the central and peripheral nervous systems, manifesting as Lewy bodies. Modifications of alpha-synuclein are implicated in mechanisms of both gain-of-function and loss-of-function in the context of PD. Lewy bodies are associated with neurotoxicity and are known to instigate neuroinflammation, ultimately leading to neurodegeneration. Furthermore, the loss of normal αSyn functionality disrupts presynaptic and postsynaptic dynamics, as well as dopaminergic transmission. Given the abundance of αSyn and its multifaceted roles in the nervous system, significant efforts are being directed toward identifying effective diagnostic biomarkers for Parkinson’s disease. However, the therapeutic potential of targeting αSyn has thus far been underwhelming. This raises important questions about whether an exclusive focus on αSyn is the most effective strategy for discovering disease-modifying treatments for PD. It is likely that merely halting the pathological aggregation and accumulation of αSyn will not address other underlying mechanisms driving PD, such as mitochondrial dysfunction, lysosomal impairment, and the ongoing influence of environmental and genetic factors. Given that age is the most consistent risk factor for the development of PD, further investigation is warranted into the roles of immunosenescence and neuroinflammatory processes in this disease. Several studies have highlighted the activation of microglia and elevated cytokine levels in the central nervous system in relation to increased αSyn expression during the progression of PD. However, several questions remain unresolved. Specifically, does the immune system react to the overexpression of αSyn, leading to unnecessary neuroinflammation and resultant neurodegeneration? Alternatively, do immunosenescence processes weaken the immune defensive mechanisms, thereby rendering the system more susceptible to αSyn-induced insults in PD? Also, are there any PD immune system-specific targets that would help to modify the course of the disease? Comprehensive studies are needed to evaluate the immune responses of patients with Parkinson’s disease to provide clearer insights into these issues.

The strengths of current αSyn research lie in the development of highly sensitive diagnostic tools, such as the αSyn SAA. Early studies measured various forms of αSyn in CSF and blood, including total, oligomeric, and phosphorylated αSyn, but limitations like sample contamination and variability between labs hindered consistent results [132]. Technological advancements like seed amplification assays (SAAs) have provided powerful tools to detect misfolded αSyn, offering high sensitivity and specificity, particularly in CSF samples [7]. Less invasive techniques, such as skin punch biopsy and serum assays, are now being explored, and could offer more accessible and repeatable diagnostic options [140]. However, challenges remain in terms of the high cost, the need for a skilled workforce, and the lack of a quantitative biomarker to track disease progression and treatment efficacy. The need for further refinement of diagnostic assays persists, particularly in understanding variations in αSyn biomarkers across genetic forms of PD. Future research should aim to standardize methodologies and further investigate how different assays can be applied across various clinical stages and patient populations.

The ability to detect misfolded αSyn in biological fluids has brought us closer to identifying early biomarkers for PD, which is critical for initiating treatment in the prodromal stages. Current treatments remain focused on symptomatic relief, and the lack of effective biomarkers for early diagnosis further complicates efforts to intervene before significant neuronal damage has occurred. Therapeutic strategies aimed at reducing αSyn aggregation or enhancing its clearance through autophagy may offer novel treatments that could slow or stop disease progression. Furthermore, personalized medicine approaches targeting specific genetic mutations or αSyn conformations could provide more tailored and effective therapies for PD patients. Current therapeutic targets include αSyn, neuroinflammation, mitochondrial function, and genetic mutations like GBA and LRRK2 [158,159,160]. Several approaches are under investigation, including reducing αSyn expression via RNA interference, inhibiting fibrillation with small molecules, and enhancing its clearance through autophagy or proteasomal pathways. Immunotherapy targeting αSyn is also a promising option, with both passive and active immunization strategies being explored. Passive immunization involves monoclonal antibodies targeting αSyn to prevent cell-to-cell transmission. However, early phase II clinical trials, such as those for Cinpanemab and Prasinezumab, have faced challenges, showing limited efficacy [161,162]. The task of delivering therapies across the blood–brain barrier is a major technical hurdle and questions remain how well current measures capture therapeutic effects. This review further highlights the need for longer trial periods, improved outcome measures, and more effective treatment delivery methods.

The role of αSyn in PD pathogenesis is intricate, encompassing both its physiological functions and its pathological aggregation. The development of reliable biomarkers for early diagnosis, alongside advances in immunotherapy, holds great promise for the future of PD treatment. However, significant obstacles remain, particularly in understanding the full spectrum of αSyn’s roles and in translating preclinical successes into effective clinical therapies. This review extends previous works by integrating the latest findings on αSyn biology and offering new insights into its potential as both a biomarker and a therapeutic target. Continued research in this area will be essential for unlocking new strategies to combat PD and related neurodegenerative disorders.

## 12. Conclusions

Thus, to summarize, the field of αSyn research has made a great advancement in recent years and many studies show that αSyn has a very important role in the pathogenesis of PD since it is implicated in a great variety of mechanisms proposed to lead to neurodegeneration. Notably, the primary loss-of-function, as well as gain-of-function mechanisms of αSyn are important and still not well understood. The research into PD biomarkers based on the pathogenic mechanism has shown promising results. Conversely, the incidental findings of αSyn-related pathology in the brains of aged individuals who had no Parkinsonism raise questions whether αSyn alone is sufficient to cause PD or other neurodegenerative diseases. A quantitative biomarker to measure disease progression and possible treatment efficacy is not available. However, current biomarkers exert a high economical cost and require a highly skilled workforce, which hampers their availability worldwide. A caveat remains for anti-αSyn therapy, and that is the need for an improved understanding of the physiological function of αSyn and how it becomes disrupted to evoke neurotoxicity and neuropathology, since this will help direct therapeutic approaches. Correspondingly, this suggests that further goalposts remain and targeting only αSyn might not be sufficient to seek biomarker and disease-modifying therapy in the near future.

## Figures and Tables

**Figure 1 biomedicines-12-02121-f001:**
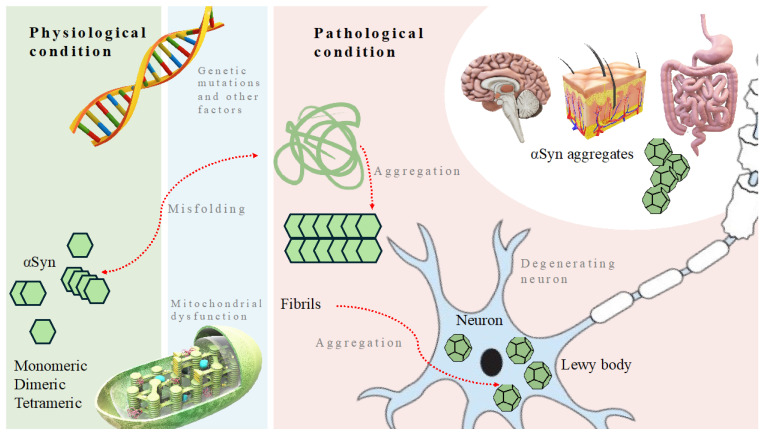
Molecular mechanisms contributing to Parkinson’s disease. Genetic mutations, mitochondrial dysfunction, and other significant biological components negatively influence physiological condition and lead to misfolding and aggregation of α-synuclein, which contributes to formation of aggregates found in the brain, skin, gut, and other organs.

**Figure 2 biomedicines-12-02121-f002:**
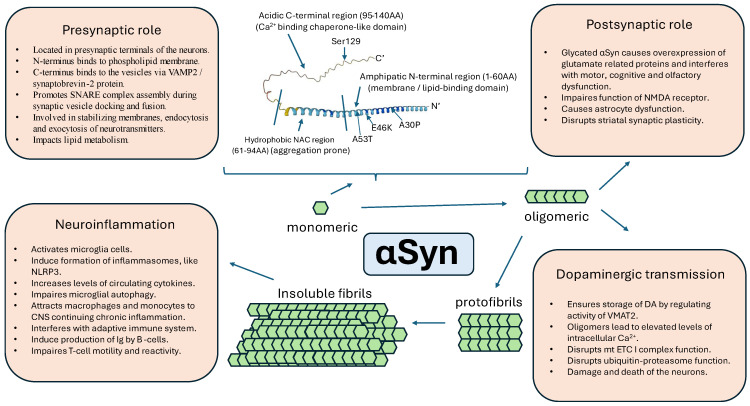
Overview of the proposed functions of physiological and pathological forms of alpha-synuclein. VAMP2—vesicle associated membrane protein 2, SNARE—soluble N-ethylmaleimide sensitive factor, NAC—non-Aβ-amyloid component, Ser—serine, AA—amino acids, α-syn—alpha-synuclein, DA—dopamine, mt—mitochondria, ETC—electron transport chain, NMDA—N-methyl-D-aspartate, NLRP3—nucleotide-binding oligomerization domain-leucine-rich repeat-pyrin domain-containing 3.

**Table 1 biomedicines-12-02121-t001:** Key characteristics from αSyn SAA studies.

Authors	Pathology Groups (N)	Sample Tissue	Assay	Sensitivity, %	Specificity, %
Manne et al., 2020 [142]	PD (13)	SG	RT-QuIC	100	94.0
Stefani et al., 2021 [143]	PD (41) + IRBD (63)	OM	RT-QuIC	45.2	89.8
Okuzumi et al., 2023 [9]	PD (275)	Blood serum	IP/RT-QuIC	94.6	92.1
Fenyi et al., 2019 [144]	PD (18)	GI	PMCA	55.6	90.9
Vascellari et al., 2023 [145]	PD (27)	GI	RT-QuICR	95.7	100
Luan et al., 2022 [146]	PD (75)	Saliva	RT-QuIC	76.0	94.4
Vivacqua et al., 2023 [147]	PD (37)	Saliva	RT-QuIC	83.8	82.6
Luan et al. 2024 [148]	PD (101)	Saliva	RT-QuIC	70.3	92.5
Russo et al., 2021 [149]	PD (30)	CSF	RT-QuIC	86.0	97.0
Siderowf et al., 2023 [8]	PD (545)	CSF	RT-QuIC	87.7	96.3
Concha-Marambio et al., 2023 [150]	PD (74)	CSF	RT-QuIC	94.6	98.0
Wang et al., 2021 [151]	PD (47)	Skin	RT-QuIC	94.0	98.0
Kuzkina et al., 2021 [152]	PD (34)	Skin	RT-QuiC	90.9	86.7
Li et al., 2024 [153]	PD (30)	Skin	RT-QuIC	93.3	100
Iranzo et al., 2021 [154]	IRBD (52)	CSF	RT-QuIC	90.4	90.0

PD—Parkinson’s disease; SG—submandibular gland; RT-QuIC—real-time quaking-induced conversion; IRBD—idiopathic rapid eye movement sleep behavior disorder; OM—olfactory mucosa; IP/RT-QuIC—immunoprecipitation-based real-time quaking-induced conversion; GI—gastrointestinal tract, PMCA—protein misfolding cyclic amplification; RT-QuICR—rapid αSyn RT-QuIC assay; CSF—cerebrospinal fluid.

## Data Availability

No new data were created or analyzed in this study.

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
