# Peer review of "Navigating the Neurobiology of Parkinson’s: The Impact and Potential of α-Synuclein"

_biomedicines, 2024, doi:10.3390/biomedicines12092121_

Round 1

Reviewer 1 Report

Comments and Suggestions for Authors

This review provides a comprehensive overview of the role of αSyn in PD, including its potential applications in pathology, biomarker development, and therapeutic strategies. Here are some of my review comments on this article:

1. The introduction section already provides a good overview of the importance of Parkinson's disease and its current research status, but the persuasive power of the introduction could be enhanced by more specific references to some of the major discoveries or research results in recent years. In addition, it is recommended that the main goal and contribution of this paper, which is to deeply analyze the role of αSyn in the pathogenesis of Parkinson's disease and to explore its potential as a biomarker and therapeutic target, be more clearly stated in the introduction.

2. There is a large of literature and reviews on alpha-synuclein and Parkinson's disease. In this review, it is recommended that the authors try to keep up with the latest developments. Most of the reviews cited in the article are five years old or even older, and it is recommended that references to the latest findings be added to enhance the significance, timeliness, and authority of the article.

3. The effects of several pathologic mechanisms (e.g., genetic mutations, mitochondrial dysfunction, etc.) on αSyn aggregation mentioned in the text could be further elaborated with specific pathways of action and recent findings for each mechanism.

4. The article mentions more extensive data on the sensitivity and specificity of studies and biomarkers of α-synuclein aggregation, but suggests a more detailed description of the data sources and the addition of graphs to make it easier for readers to understand.

Overall, The article has important academic value in the research field of Parkinson's disease, but there is still room for improvement in terms of literature update and data support. It is hoped that the authors will further improve the article after revision.

Author Response

Comment 1: The introduction section already provides a good overview of the importance of Parkinson's disease and its current research status, but the persuasive power of the introduction could be enhanced by more specific references to some of the major discoveries or research results in recent years. In addition, it is recommended that the main goal and contribution of this paper, which is to deeply analyze the role of αSyn in the pathogenesis of Parkinson's disease and to explore its potential as a biomarker and therapeutic target, be more clearly stated in the introduction.

Response 1: Thank you for the comment and truthful insight. Therefore, in the introduction we have included Okuzami and colleagues’ breakthrough achievement with blood-based assay targeting αSyn in serum, which is one of the research projects helping to believe that we could confirm a diagnosis according to a blood test targeting αSyn in serum. In addition, we divided our goal into different points, believing that it will help our reader quickly understand the main aim and central point of this article.

These changes can be found on p. 2, paragraph 1, 83-89 and 94-100 line.

Comment 2: There is a large of literature and reviews on alpha-synuclein and Parkinson's disease. In this review, it is recommended that the authors try to keep up with the latest developments. Most of the reviews cited in the article are five years old or even older, and it is recommended that references to the latest findings be added to enhance the significance, timeliness, and authority of the article.

Response 2: Thank you, we updated the list of references by including the latest articles.

Comment 3: The effects of several pathologic mechanisms (e.g., genetic mutations, mitochondrial dysfunction, etc.) on αSyn aggregation mentioned in the text could be further elaborated with specific pathways of action and recent findings for each mechanism.

Response 3: Thank you, useful observation. We added more specific information related to pathogenesis and genetic mutations as well as mitochondrial dysfunction.

These changes can be found on p. 6, paragraph 1, 253-256 and 259-264 line.

Comment 4: The article mentions more extensive data on the sensitivity and specificity of studies and biomarkers of α-synuclein aggregation, but suggests a more detailed description of the data sources and the addition of graphs to make it easier for readers to understand.

Response 4: Thank you for pointing this out. In response to your comment, we have created a table that consolidates the relevant studies and their corresponding results, including sample tissue, sensitivity and specificity values. This table is intended to make it easier for readers to analyze and compare the data.

The changes made can be found on p. 13-14, table 1.

Reviewer 2 Report

Comments and Suggestions for Authors

In this paper, the author reviewed the role of α-Synuclein in Parkinson's Pathogenesis. It is interesting, but there are some shortcomings.

1.     In Part 10, it is suggested that the author use a table to list the targets, treatment strategies, clinical progress, and other information of PD treatment for alph synuclein, in order to improve the readability of the article.

2.     In the conclusion section, it is suggested that the author supplement future strategies for PD treatment targeting alpha synuclein.

3.     In the "Main aspects of the review" section, some sentences have incomplete meanings, such as, large scale studies already showed high pooled sensitivity of 91% and specificity of 95% when distinguishing PD patients from healthy controls. It is suggested that the author reorganize the description and logical relationship of this section.

Comments on the Quality of English Language

 In the "Main aspects of the review" section, some sentences have incomplete meanings, such as, large scale studies already showed high pooled sensitivity of 91% and specificity of 95% when distinguishing PD patients from healthy controls. It is suggested that the author reorganize the description and logical relationship of this section.

Author Response

Comment 1: In Part 10, it is suggested that the author use a table to list the targets, treatment strategies, clinical progress, and other information of PD treatment for alph synuclein, in order to improve the readability of the article.

Response 1: Thank you for your note. In this part of the article, we have only briefly mentioned the treatment strategies which are under investigation such as small molecular inhibitor, enhancement of protein clearance and gene therapy. We thought that the central view should be pointed to immunotherapy, especially passive immunization, because they are in a Phase II, that’s why we do not expand to other therapies and their strategies, different targets, clinical progress. Moreover, other therapies do not reflect our provided information during the text, so the table providing more information would be more away from the main point.

Comment 2: In the conclusion section, it is suggested that the author supplement future strategies for PD treatment targeting alpha synuclein.

Response 2: Thank you for pointing this out. We agree with this comment. In response to your comment, we have included some thoughts about future therapies targeting αSyn.

These changes can be found on p. 16, paragraph 1, 732-742 line.

Comment 3: In the "Main aspects of the review" section, some sentences have incomplete meanings, such as, large scale studies already showed high pooled sensitivity of 91% and specificity of 95% when distinguishing PD patients from healthy controls. It is suggested that the author reorganize the description and logical relationship of this section.

Response 3: Thank you for the insight. In response to your comment, we have carefully reorganized the sentences and refined the logical flow of the section. Specifically, we have restructured the sentences to ensure that the meaning is complete and the relationships between the data points and conclusions are clear. The revised section now presents the findings from large-scale studies in a more coherent and logically structured manner, improving the overall readability and clarity of the discussion.

The changes made can be found on p. 1, 42-45 line

Reviewer 3 Report

Comments and Suggestions for Authors

Authors gave a very good and comprehensive review about alpha-synuclein and PD pathology. It is very helpful in the field.

Authors have covered the alpha-synuclein normal function, its pathological changes, proteinopathy-proteinopenia, clinical aspects as well as alpha synuclein as a potential therapeutic target for PD…

One of pathophysiological features of neurodegenerative disorders is that aggregated alpha-synuclein not only presents in the brain of PD patients, but also presents in other brains of neurodegenerative disorders. It always co-exists with many other neurotoxic proteins such as Abeta, p-tau.

Is it possible that authors discuss about this phenomenon?

Author Response

Comment 1: One of pathophysiological features of neurodegenerative disorders is that aggregated alpha-synuclein not only presents in the brain of PD patients, but also presents in other brains of neurodegenerative disorders. It always co-exists with many other neurotoxic proteins such as Abeta, p-tau. Is it possible that authors discuss about this phenomenon?

Response 1: Thank you for your remarks. We agree with you that there is some overlap and similarities between synucleinopathies and tauopathies. Despite their differences in structure, cellular localization and primary associations with specific disorders, the proteins share same similarities like protein aggregation, pathological spreading and role in synaptic dysfunction. Although tau pathology can be observed in PD and Lewy body pathology in AD, this co-occurrence suggests some overlapping or synergistic mechanisms in neurodegneration, but these aspects weren’t the central point of our article and we thought that reviewing other neurodegenerative disorders would require more volume of the article and that would be changeful to the main goals, so we decided to exclude this part.

Round 2

Reviewer 1 Report

Comments and Suggestions for Authors

The authors have answered the reviewers' questions and provided reasonable explanations and appropriate revisions.

Author Response

The authors have answered the reviewers' questions and provided reasonable explanations and appropriate revisions.

Response: Thank you.

Reviewer 2 Report

Comments and Suggestions for Authors

None

Author Response

None